# Effect of Cavitation Peening on Fatigue Properties in Friction Stir Welded Aluminum Alloy AA5754



**Hitoshi Soyama** [1,*], **Michela Simoncini** [2] **and Marcello Cabibbo** [3]

1 Department of Finemechanics, Tohoku University, Sendai 980-8579, Japan
2 Faculty of Engineering, e-Campus University, Via Isimbardi 10, 22060 Novedrate, Italy; michela.simoncini@uniecampus.it
3 Department of Industrial Engineering and Mathematics, Università Politecnica delle Marche, Via Brecce Bianche, 60131 Ancona, Italy; m.cabibbo@staff.univpm.it
* Correspondence: soyama@mm.mech.tohoku.ac.jp; Tel.: +81-22-795-6891; Fax: +81-22-795-3758

**Abstract:** Friction stir welding (FSW) is an attractive solid-state joining technique for lightweight metals; however, fatigue properties of FSWed metals are lower than those of bulk metals. A novel mechanical surface treatment using cavitation impact, i.e., cavitation peening, can improve fatigue life and strength by introducing compressive residual stress into the FSWed part. To demonstrate the enhancement of fatigue properties of FSWed metal sheet by cavitation peening, aluminum alloy AA5754 sheet jointed by FSW was treated by cavitation peening using cavitating jet in air and water and tested by a plane bending fatigue test. The surface residual stress of the FSWed part was also evaluated by an X-ray diffraction method. It was concluded that the fatigue life and strength of FSWed specimen were improved by cavitation peening. Whereas the fatigue life at $\sigma_a$ = 150 MPa of FSWed specimen was about 1/20 of the bulk sheet, cavitation peening was able to extend the fatigue life of the non-peened FSW specimen by 3.6 times by introducing compressive residual stress into the FSWed part. This is the first paper to demonstrate the improvement of fatigue properties of FSWed metallic sheet by cavitation peening.

**Keywords:** surface treatment; cavitation peening; friction stir welding; fatigue life; aluminum alloy



## 1. Introduction

Friction stir welding (FSW) was invented at The Welding Institute (Cambridge, UK) [1,2]. FSW is an attractive solid-state welding technology for light-weight metals [3–7] in transport industries [8], such as aerospace [9] and automotive [10]. FSW is an ecological and green technology, as it can reduce material waste and avoid harmful gas emissions due to conventional welding [11,12]. However, the fatigue life and strength of conventional FSWed metal is considerably lower than those of base metal [13–19]. Thus, it is worthwhile to develop a method to improve fatigue properties of FSWed metals.

As is well known, the fatigue properties strongly depend on residual stress [20]. As FSW causes stirring and a thermo-mechanically affected zone, residual stress was introduced, and tensile residual stress near the welded zone was reported [21–25]. It was reported that the residual stress of the welded side was tension and that of root side was compression [26], and a prediction method of residual stress distribution in FSW was proposed [27]. However, the measurement of residual stress is still required, as FSWed residual stress varies greatly depending on FSW conditions and materials.

A popular method to improve tensile residual stress to compression is shot peening. It was reported that shot peening and laser peening could introduce compressive residual stress into the FSWed part [28–32]. Shot peening and laser peening are referred to as mechanical surface treatment, as local plastic deformation introduced by impact is used to introduce compressive residual stress and work hardening to target material surfaces [33]. At shot peening, shot impacts, i.e., solid collisions, are used to produce the

plastic deformations, and improved fatigue strength of FSWed aluminum by shot peening has been reported [34,35]. In the case of ultrasonic peening, shots or needles accelerated by ultrasonic vibration introduce plastic deformation, and it was reported that ultrasonic peening enhanced the fatigue properties of FSWed AA7075 [36]. It was also reported that surface mechanical rolling treatment [37] and deep rolling [38] improved fatigue strength of FSWed AA7075-T651.

There are three types of laser peening. One is the water film type, in which a water film is made on the target, which is normally covered with a sacrificed layer, i.e., tape or paint, and a laser pulse on the order of nanoseconds irradiates to the target. The water film type of laser peening improved the fatigue strength of FSWed AA7075-T7351 [34]. Another type of laser peening is the submerged type. With submerged laser peening, the target is placed in water is exposed to a laser pulse on the order of nano-seconds. Note that the sacrificed layer is not required at submerged laser peening. It was reported that submerged laser peening improved the fatigue performance of FSWed A6061-T6 [39]. The third type of laser peening, developed by T. Sano et al. [40], is performed under atmospheric conditions, without a sacrificed overlay, using a femtosecond pulse laser, and it is called femtosecond laser peening. It was reported that femtosecond laser peening improved fatigue strength of FSWed 7075-T73 aluminum alloys [41]. With these three types of laser peening, laser ablation impact is used; these are a kind of shotless peening, as shot is not used. Even submerged laser peening is believed to be a laser ablation process, as the amplitude of the pressure wave in water with laser ablation was larger than that of bubble collapse [42]. Note that a bubble is generated after laser ablation in submerged laser peening, and the bubble behavior is similar to a cavitation bubble, as the bubble is developed, shrunk, and then collapsed, producing an impact [42,43]. Thus, the bubble is referred to as laser cavitation in the present paper. When an impact passing through metallic material was measured by a polyvinylidene fluoride (PVDF) sensor, which was developed to measure cavitation impacts [44,45], the impact produced by laser cavitation in submerged laser peening was larger than that of laser ablation [43]. Cavitation impact can be utilized for mechanical surface treatment, and peening using cavitation impacts are referred to as cavitation peening [46,47]. Cavitation peening might be applicable to improve the fatigue properties of FSWed metals.

In conventional cavitation peening, a high-speed submerged water jet with cavitation, i.e., a cavitating jet, is used to generate cavitation [48]. Note that cavitation peening is different from water jet peening [49], in which droplet impacts of a jet core are used. As the cavitation number is defined by the injection pressure of a nozzle and downstream pressure of the nozzle, and is the main parameter of a cavitating jet [48], a classification map using the cavitation number and distance from the nozzle to the target was proposed [33]. It has been reported that the fatigue strength of stainless steel treated by cavitation peening is better than that of shot peening, water jet peening, and submerged laser peening [43,50]. As mentioned above, a high-speed submerged water jet, i.e., a cavitating jet in water, has been used for conventional cavitation peening [47]. Moreover, a cavitating jet in air was successfully realized by injecting a high-speed water jet into a low-speed water jet, which was injected into air without a water-filled chamber [51,52], and an improvement in fatigue strength of stainless steel using a cavitating jet in air has been reported [53]. It was also reported that the cavitating jet in air can introduce large compressive residual stress at the surface, but the introduced layer is shallow [46]. On the other hand, the cavitating jet in water can introduce compressive residual stress into a deeper region, but the surface roughness is larger than that of the cavitating jet in air [46].

In order to demonstrate the improvement of fatigue properties of FSWed metals by cavitation peening, FSWed aluminum alloy AA5754 was treated by the cavitating jet in air and water, and then tested by a displacement-controlled plane bending fatigue test. To clarify the effect of residual stress on the fatigue properties of FSWed AA5754, surface residual stress was measured with a 2D method using X-ray diffraction.

## 2. Materials and Methods

### 2.1. Friction Stir Welded Aluminum Alloy AA5754

Fatigue specimens were made by milling AA5754-H114 sheets from FSWed aluminum alloy as shown in Figure 1. Figure 2 reveals the geometry of fatigue specimens and coordinates of residual stress measurement. Sheets were welded by FSW using an H13 tool steel pin with a shoulder diameter of 12 mm, a truncated conical shape with a base of 3.5 mm and a height of 1.8 mm. During FSW, the tool was initially forced 0.2 mm down into the aluminum alloy sheet. Welding was carried out with a pin nutting angle of 2°. Two different welding parameters (rotational speed, $\omega$, and welding translational speed, $v$) were used. One welding procedure was carried out with $\omega$ = 2500 rpm and $v$ = 60 mm/min, a second welding was carried out by $\omega$ = 1500 rpm and $v$ = 60 mm/min welding settings. These settings were chosen according to the weldability of the AA5754 alloy that was determined in previously published works by some of the present authors [54,55]. Ultimate tensile strength and ultimate elongation were 220 MPa and 34% for bulk metal AA5754, 222.00 MPa and 32.8% for FSWed AA5754 at $\omega$ = 1500 rpm, and 197.31 MPa and 19.30 MPa for FSWed AA5754 at $\omega$ = 2500 rpm, respectively [54].

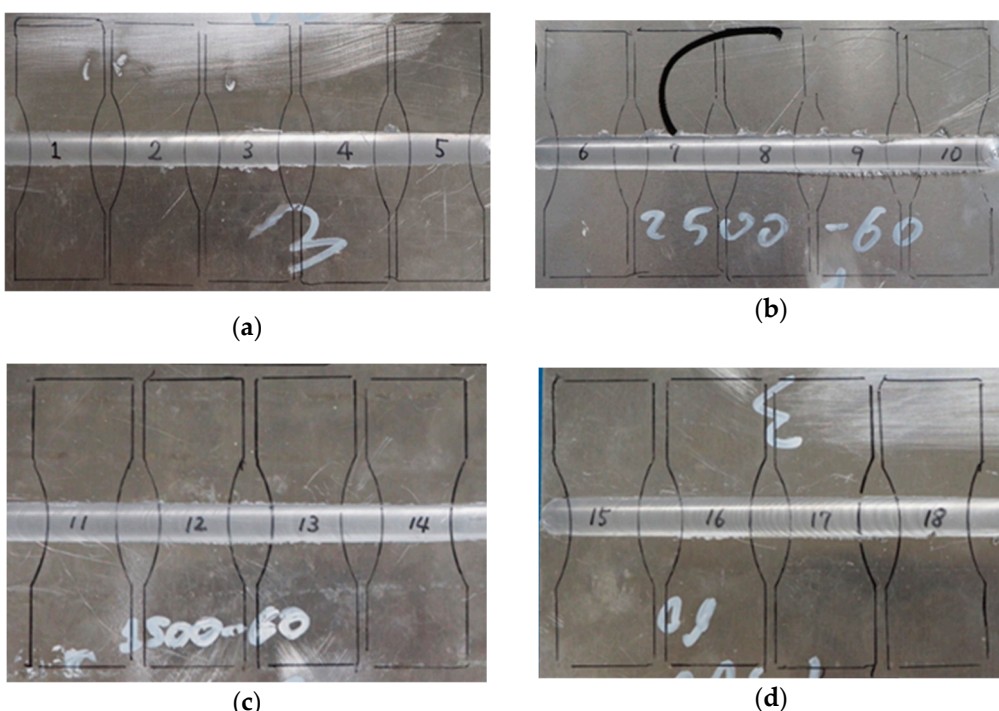

**Figure 1.** Friction stir welded (FSWed) aluminum alloy AA5754 sheets for fatigue test specimens. The black line indicates the shape of the test specimen. (**a,b**): Rotational speed $\omega$ was 2500 rpm, and transverse speed $v$ was 60 mm/min. (**c,d**): Rotational speed $\omega$ was 1500 rpm, and transverse speed $v$ was 60 mm/min.

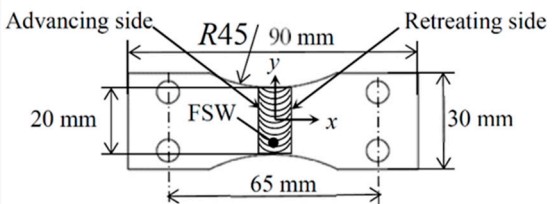

**Figure 2.** Geometry of the fatigue specimen for displacement-controlled plane-bending fatigue test and coordinates for residual stress measurement. The thickness was 2 mm.

### 2.2. Cavitation Peening Using a Cavitating Jet in Air and Water

Figure 3 illustrates the cavitation peening system using (a) cavitating jet in air (CJA) and (b) cavitating jet in water (CJW). In the case of CJA, a high-speed water jet was pressurized by a plunger pump, and it was injected into a low-speed water jet, which was pressurized by a submerged turbine pump, using a concentric nozzle. In the present experiment, the CJA's conditions were as follows, considering results of the previous reports [52,56]. The throat diameter of the nozzle for the high-speed water was 1 mm; the inner diameter for the low-speed water was 30 mm; the injection pressures of the high-speed water jet and the low-speed water jet were 20 MPa and 0.05 MPa, respectively; the standoff distance from the nozzle to the target was 56 mm. The scanning time per unit length, $t_p$, was defined by the following equation:

$$t_p = \frac{n}{v} \tag{1}$$

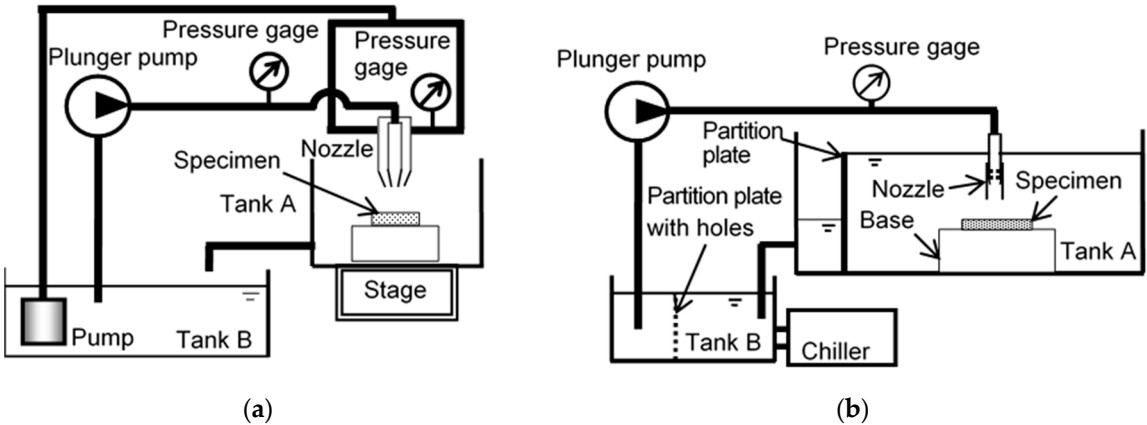

**Figure 3.** Schematic diagram of cavitation peening systems. (**a**) Cavitating jet in air. (**b**) Cavitating jet in water.

Here, $n$ is the number of scan and $v$ is scanning speed, i.e., relative speed between the nozzle and the specimen. At the present experiment of CJA, $n = 1$, $v = 4$ mm/s, $t_p = 0.25$ s/mm was chosen, as the maximum speed of the cavitation peening system using CJA, which was developed for the treatment of tool steel alloy [51], was $v = 4$ mm/s.

In the case of CJW, the high-speed water jet was injected into the water-filled tank A by using the plunger pump through a nozzle. In order to enhance the aggressive intensity of the cavitating jet in water, a cavitator and a guide pipe were installed in the nozzle [57]. The outlet geometry of the nozzle was also optimized considering the previous report [58]. The nozzle throat diameter $d$ was 2 mm, and the standoff distance $s$ was 222 mm, as the peening intensity had a peak at $s = 222$ mm [43]. The injection pressure $p_1$ was 30 MPa. Note that cavitation number $\sigma$, which is a key factor of the cavitating flow, was defined by Equation (2):

$$\sigma = \frac{p_2 - p_v}{p_1 - p_2} \approx \frac{p_2}{p_1} \tag{2}$$

Here, $p_2$ and $p_v$ are downstream pressure of the nozzle and the vapor pressure of the water, respectively. As $p_1 \gg p_2 \gg p_v$, $\sigma$ is simplified to Equation (2). According to a previous report [33], when $s/d$ satisfies Equation (3), the cavitation peening condition is met:

$$\frac{s}{d} > 1.8\, \sigma^{-0.6} \tag{3}$$

In the present condition, $s/d = 111$ and $1.8\, \sigma^{-0.6} = 55.1$, as $\sigma = 0.0033$, then the used condition was cavitation peening condition. In the present experiment of CJW, $n = 2$,

$v$ = 1 mm/s, $t_p$ = 2 s/mm was chosen, as the density of the plastic deformation pits induced by CJW was lower than that of CJA.

### 2.3. Evaluation of Fatigue Properties and Surface Characteristics

Fatigue properties of FSWed AA5754 were evaluated using a Schenk-type displacement-controlled plane-bending fatigue test. [59]. The span length at fixed point was 65 mm as shown in Figure 2. The test frequency was 12 Hz. As the displacement-controlled fatigue tester (Tokyo Koki Co. Ltd., Tokyo, Japan) was used, the amplitude of bending stress could not be fixed exactly, and the number of cycles of FSWed specimen $N_{f150}$ at $\sigma_a$ = 150 MPa was obtained by the following procedure. Note that *S-N* line of base metal and FSWed specimen were straight at $\sigma_a$ = 150 MPa as mentioned later. Then, $\sigma_a$ = 150 MPa was chosen to discuss the fatigue life at the present experiment. It was assumed that the *S–N* curve for low-cycle-fatigue base metal (BM) specimens is described by Equation (4) and that that for FSWed specimens is described by Equation (5), where $c_1$, $c_2$, and $c_3$ are constants. Namely, these curves are parallel to each other, as shown in Figure 4.

$$\sigma_{a\ BM} = -c_1 \log N_{fBM} + c_2 \tag{4}$$

$$\sigma_{a\ FSW} = -c_1 \log N_{fFSW} + c_3 \tag{5}$$

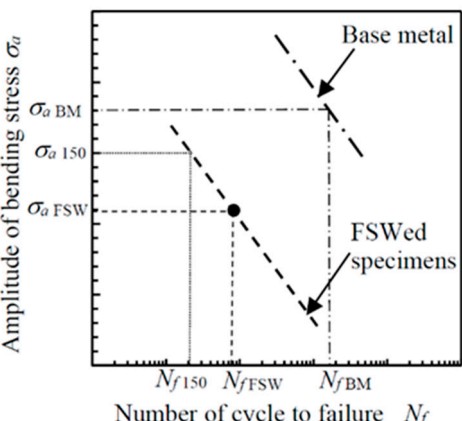

**Figure 4.** Schematics to obtain $N_{f150}$ of FSWed specimen using the *S-N* curve of the base metal (BM) specimen.

In the present experiment, $c_1$ and $c_2$ were obtained from 8 experimental datasets of base metal specimens using a least square method. The number of fatigue specimens for base metal and FSWed sheet was 8 and 17, respectively. The $c_3$ of FSWed specimen at each condition was obtained from $c_1$ and experimental dataset of $\sigma_{a\ FSW}$ and $N_{f\ FSW}$ of FSWed specimens. Therefore, $N_{f\ 150}$ for FSWed specimens is given by Equation (6):

$$\sigma_{a150} = -c_1 \log N_{f150} + c_3 \tag{6}$$

From Equation (6), we obtain:

$$N_{f150} = 10^{\frac{c_3 - \sigma_{a150}}{c_1}} \tag{7}$$

From this, we obtain $N_{f\ 150}$. The $\sigma_a$ at $N_f$ = $10^5$ is also obtained by Equation (8):

$$\sigma_a = -c_1 \log 10^5 + c_3 \tag{8}$$

As mentioned in the introduction, residual stress is one of the key factors of the fatigue properties of FSWed metals, the surface residual stress was measured with a 2D method [60]

using an X-ray apparatus with a two-dimensional detector (Bruker Japan K. K., Tokyo, Japan), as the 2D method can be applied for local areas such as those on the sub-millimeter order. The X-ray used was K$\alpha$ X-ray from a Cr tube operated at 35 kV and 40 mA through a 146.0 μm diameter total reflection collimator and with an incident monochromator. The lattice plane (*h k l*) used for the surface residual stress measurement was the Al (3 1 1) plane, and the diffraction angle without strain was 139.3 degrees. The Young's modulus and Poisson ratio used were 69.3 MPa and 0.35, respectively. The 24 diffraction rings from the specimen at various angles were detected, and the exposure time per frame at each single position was 10 min.

As the fatigue properties of the peened specimen also depend on the surface roughness and the surface hardness, the arithmetic mean roughness, *Ra*, the maximum height of the roughness, *Rz*, and the Vickers hardness at the surface, *HV*, were measured. At the hardness measurement, the surface of specimen was not polished. The surface roughness was evaluated using a stylus-type profilometer (Tokyo Seimitsu Co., Ltd., Hachioji, Japan). The load used in measuring the Vickers hardness was 0.2 kgf, i.e., 1.96 N. The hardness was measured seven times in each case, and the mean value and standard deviation were obtained from five points, excluding the maximum and minimum values.

In order to observe the surface characteristics of FSWed AA5754, the specimen surface was observed by a digital microscope (Keyence Corporation, Osaka, Japan). To investigate the crack initiation point, the fractured surface of the specimen was also observed by a scanning electron microscope (JEOL Ltd., Akishima, Japan) (SEM).

## 3. Results

### 3.1. Aspect of Cavitation Peened Surface

Figure 5 reveals the aspect of the root side surface of FSWed AA5754 treated by (a) cavitating jet in air and (b) cavitating jet in water taken by the digital camera, as the fracture initiated from the root side surface in most cases. To show the plastic deformation pits produced by the cavitating jet in air, Figure 6 reveals the magnified aspect of the welded side surface of FSWed AA5754 treated by (a) cavitating jet in air and (b) cavitating jet in water, as taken by the digital microscope. As shown in Figures 5 and 6, the plastic deformation pits were introduced by the cavitating jet in air and water. The pit size generated by the cavitating jet in air was smaller than that of the cavitating jet in water, as previously reported [46]. Note that the pits introduced by the cavitating jet in air and water were plastically deformed pits without mass loss, as cavitation peening was finished within an incubation stage of cavitation erosion progress, in which mass loss did not occur [61].

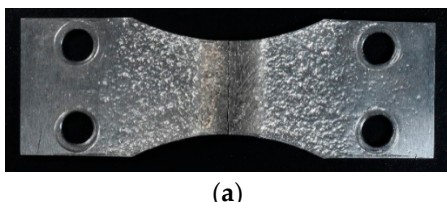 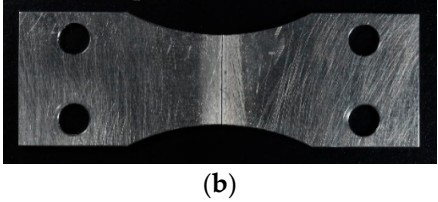

(**a**) (**b**)

**Figure 5.** Aspect of the root side surface of cavitation-peened specimens to show plastic deformation pits taken by digital camera. (**a**) Cavitating jet in air. (**b**) Cavitating jet in water.

In order to show the surface roughness of cavitation peened specimens, Figure 7 illustrates the arithmetical mean roughness *Ra* and the maximum height of roughness profile *Rz* of the base metal part of FSWed AA5754 treated by the cavitating jet in air (CJA) and the cavitating jet in water (CJW) compared with non-peened specimen NP. As shown in Figures 5 and 6, the pits introduced by CJW were larger than that of CJA, both *Ra* and *Rz* of CJW were larger than those of CJA. The reason why the pits induced by CJW were larger than those of CJA is that the nozzle throat diameter of CJW was larger than that of CJA. Namely, the cavitating jet using the larger nozzle throat causes larger pits, due to scaling effect of nozzle throat diameter [46]. Whereas the NP specimen has very low arithmetical

mean roughness compared with CJA, the number of cycles to failure of NP is less than the FSWed specimen treated by CJA. The main reason is that CJA introduces compressive residual stress. More details are described in the Sections 3.2 and 3.3.

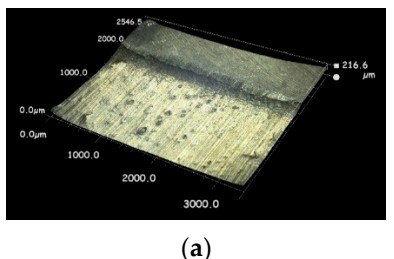

(**a**)

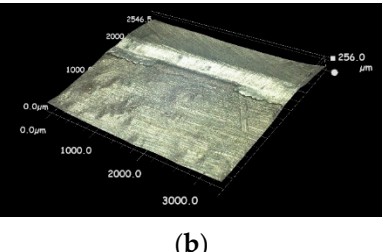

(**b**)

**Figure 6.** Aspect of the welded side surface of cavitation-peened specimens observed by digital microscope. (**a**) Cavitating jet in air. (**b**) Cavitating jet in water.

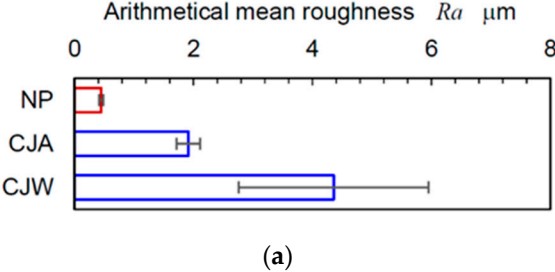

(**a**)

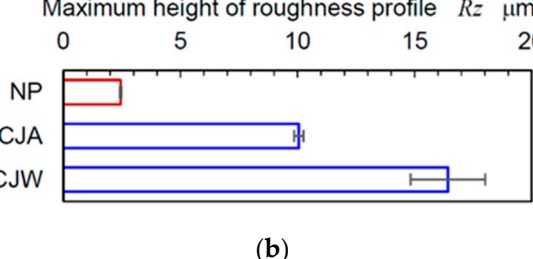

(**b**)

**Figure 7.** Surface roughness of cavitation-peened specimens by cavitating jet in air (CJA) and cavitating jet in water (CJW) comparing with non-peened specimen NP. (**a**) Arithmetical mean roughness. (**b**) Maximum height of roughness profile.

In order to investigate the work-hardening effect of FSW by cavitation peening, Figure 8 illustrates the surface Vickers hardness $HV$ of base metal (BM) part and FSWed part treated by CJA and CJW comparing with NP. The $\omega$ at FSW was 1500 rpm. Whereas the standard deviation was considerably large, the average value of the surface Vickers hardness at the BM part was slightly increased by cavitation peening. The surface Vickers hardness at the FSWed part for all three cases was decreased compared with that of the BM part. A significant work hardening effect was not observed in the present condition.

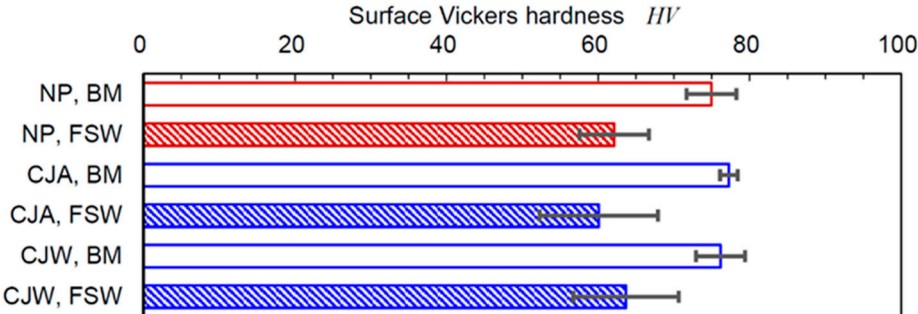

**Figure 8.** Surface hardness of base metal (BM) part and FSWed part of cavitation peened specimens by cavitating jet in air (CJA) and cavitating jet in water (CJW) compared with non-peened specimen NP.

*3.2. Improvement of Fatigue Properties of FSWed AA5754 by Cavitation Peening*

To show the effect of cavitation peening on the fatigue properties of FSWed AA5754 compared with the non-peened specimen (NP), Figure 9 illustrates the result of the plane-bending fatigue test of FSWed AA5754 treated with the cavitating jet in air (CJA) and the cavitating jet in water (CJW). The result of the base metal of AA5754 is also shown in

Figure 9. In most cases of FSWed AA5754, the fatigue started from the root side of the FSWed part, except for the two points indicated by "FSW boundary on the welded side" in Figure 9. Note that in the case indicated by "corner of root side", the crack started at the corner of the root side (see Figure 10a). For the reference, in order to show started point of fracture, schematic diagrams were revealed in Figure 9. To investigate the quantitative effect on the fatigue life and strength, Table 1 reveals $N_{f\,150}$ and $\sigma_a$ at $N_f = 10^5$ for base metal and FSWed AA5754 obtained using Equations (7) and (8). $c_1$ and $c_2$ obtained by the present experiment were 62.1 $\pm$ 10.8 and = 514.7 $\pm$ 56.5, respectively. As shown in Figure 9, the fatigue life and strength of FSWed AA5754 were considerably smaller than those of the base metal. For example, whereas $N_{f\,150}$ and $\sigma_a$ at $N_f = 10^5$ of base metal were 7.45 $\times$ 10$^5$ and 204.2 MPa, those of FSWed AA5754 without peening, excluding "corner of root side", were 3.38 $\times$ 10$^4$ and 120.7 MPa for $\omega$ = 1500 rpm and 2.10 $\times$ 10$^4$ and 107.9 MPa for $\omega$ = 2500 rpm, respectively. $N_{f\,150}$ and $\sigma_a$ at $N_f = 10^5$ of FSWed AA5754 without peening were 4.54% and 59% for $\omega$ = 1500 rpm and 2.82% and 53% for $\omega$ = 2500 rpm compared with base metal. This result suggests that FSW at $\omega$ = 1500 rpm was slightly better than that of $\omega$ = 2500 rpm at the present condition.

In the case of FSW at both $\omega$ = 1500 rpm and $\omega$ = 2500 rpm, cavitation peening improved $N_{f\,150}$ and $\sigma_a$ at $N_f = 10^5$. As shown in Figure 9, datapoints treated by CJW were scattered. This might be caused by the relatively large surface roughness as shown in Figure 7. Note that both data points of CJW in Figure 9 were used to obtain $N_{f\,150}$ and $\sigma_a$ at $N_f = 10^5$ of CJW in Table 1. In the case of CJA, $N_{f\,150}$ and $\sigma_a$ at $N_f = 10^5$ were 1.23 $\times$ 10$^5$ and 155.7 MPa for $\omega$ = 1500 rpm and 5.56 $\times$ 10$^4$ and 134.2 MPa for $\omega$ = 2500 rpm, respectively. In other words, in the case of FSW at $\omega$ = 1500 rpm, cavitation peening using CJA improved the fatigue life by 3.6 times and the fatigue strength by 1.3 times, compared with the non-peened one.

In order to investigate the crack initiation point during the plane bending fatigue test, Figure 10 shows the aspect of the fractured surface observed by SEM. In some cases, which are indicated by "corner of root side" in Figure 9, the crack initiated at the corner of the root side, as shown in Figure 10a. As mentioned above, in most cases, the crack started from the flat part of the root side, as shown in Figure 10b–f. In the case of non-peened specimens, as shown in Figure 10b,c, the fatigue failure started at the surface of the root side. On the other hand, in the case of specimens treated by cavitation peening, the fatigue crack initiated at the under the surface, as shown in Figure 10d,f. In the present experiment, the fatigue properties were investigated with the plane-bending fatigue test. Thus, the maximum tensile stress was applied at the surface. After cavitation peening, if the compressive residual stress is introduced at the surface, then the maximum tensile stress of the peened specimen would be generated under the surface. Thus, the surface residual stress before and after peening is discussed in Section 3.3, "Introduction of Compressive Residual Stress by Cavitation Peening".

When metals are exposed to cavitation for a long time, metals are corroded. However, it was reported that a cavitating jet improved the corrosion resistance of a carbon steel surface [62] and the hydrodynamic cavitation produced radicals [63]. Thus, cavitation peening using a cavitating jet in air (CJA) and a cavitation jet in water (CJW) might produce a passivation layer on the aluminum alloy. The investigation of effect of an oxide layer on aluminum alloy produced by CJA and CJW is a future work.

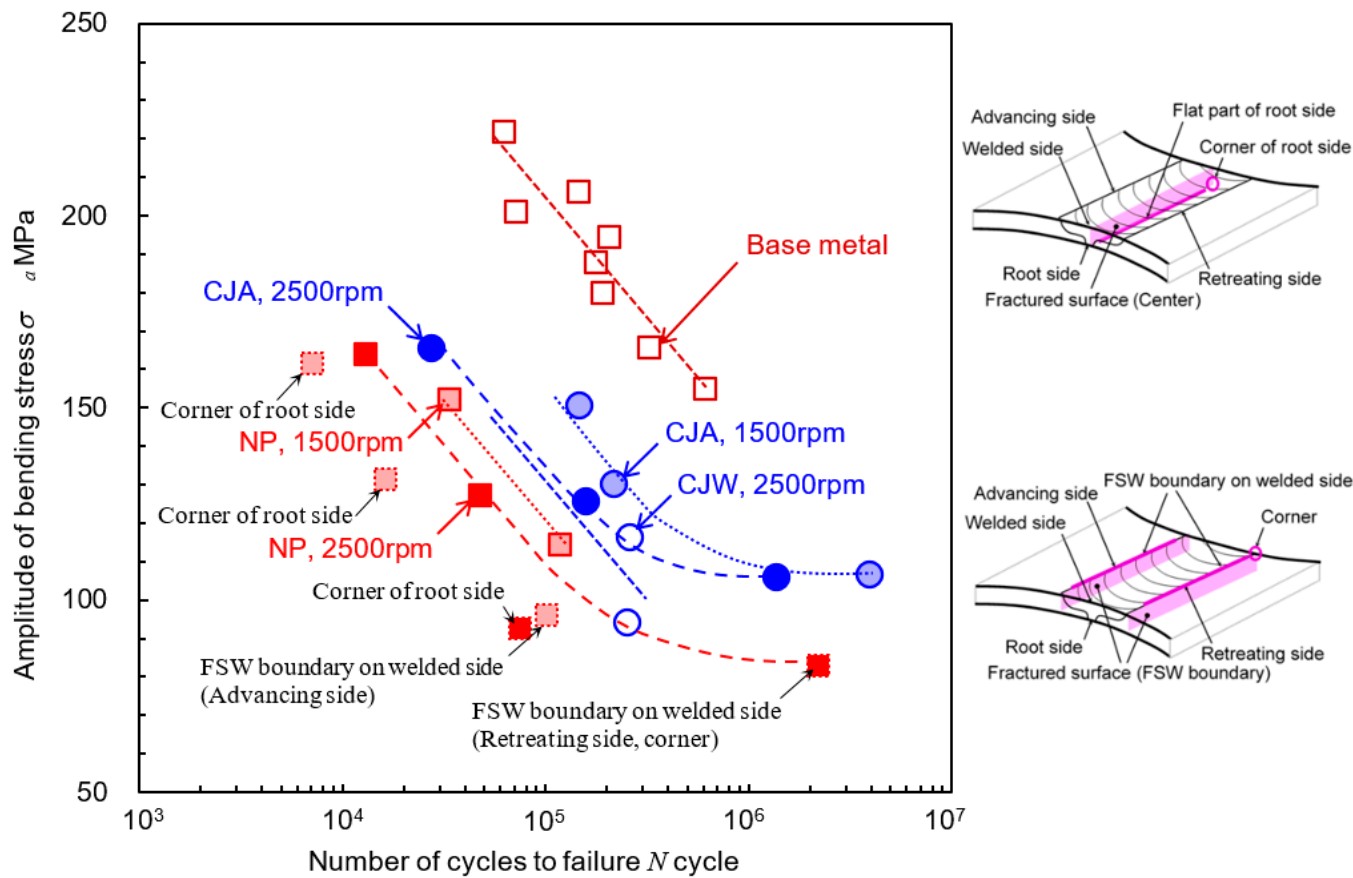

**Figure 9.** *S-N* curve obtained by plane-bending fatigue test for the base metal and FSWed specimens treated by cavitating jet in air (CJA) and cavitating jet in water (CJW) and without peening (NP). The crack initiation point of the FSWed specimen was flat part of root side, except specimens indicated by "corner of root side" and "FSW boundary of welded side".

**Table 1.** Fatigue properties of FSWed specimen treated by cavitation peening. $N_f$ at $\sigma_a$ = 150 MPa, i.e., $N_{f\,150}$, and $\sigma_a$ at $N_f = 10^5$ are obtained by Equations (7) and (8) using experimental data. CJA and CJW are cavitation peening by using cavitating jet in air and water, NP means non-peened one.

| Specimen | Peening | $c_2$ or $c_3$ | $N_{f150}$ (Cycle) | (%) | $\sigma_a$ at $N_f = 10^5$ (MPa) |
|---|---|---|---|---|---|
| Base metal | NP | 514.7 | $7.45 \times 10^5$ | (100%) | 204.2 |
| FSW ($\omega$ = 1500 rpm) | NP | 431.2 | $3.38 \times 10^4$ | (4.54%) | 120.7 |
| | CJA | 466.2 | $1.23 \times 10^5$ | (16.51%) | 155.7 |
| FSW ($\omega$ = 2500 rpm) | NP | 418.6 | $2.10 \times 10^4$ | (2.82%) | 107.9 |
| | CJA | 444.7 | $5.56 \times 10^4$ | (7.46%) | 134.2 |
| | CJW | 440.9 | $4.83 \times 10^4$ | (6.48%) | 130.4 |

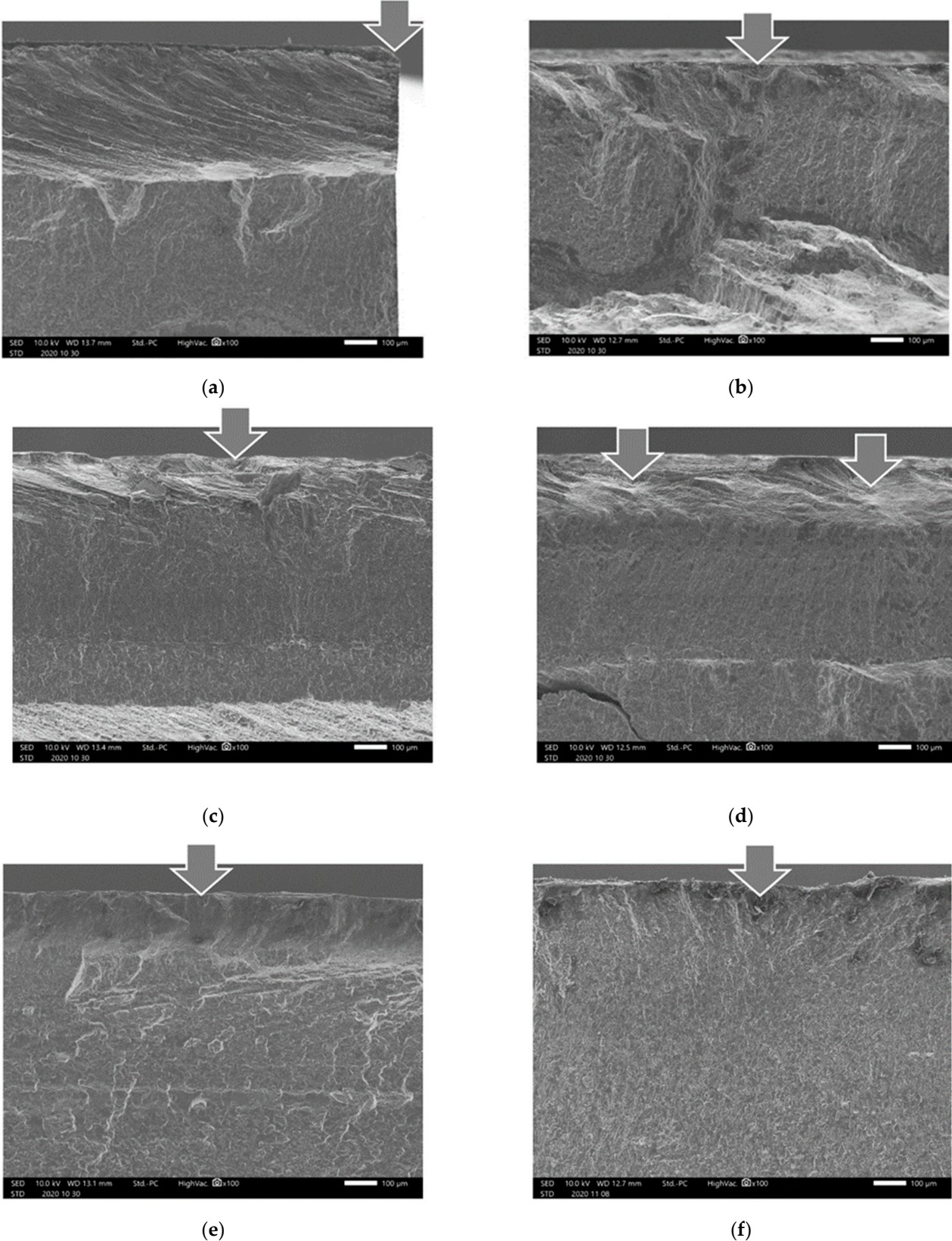

**Figure 10.** Aspect of fractured surface observed by scanning electron scope SEM. The upper surface in the figure is the root side surface. (**a**) $\omega$ = 1500 rpm, NP, $\sigma_a$ = 131 MPa, $N_f$ = 1.63 $\times$ 10⁴. (**b**) $\omega$ = 1500 rpm, NP, $\sigma_a$ = 115 MPa, $N_f$ = 1.18 $\times$ 10⁵. (**c**) $\omega$ = 2500 rpm, NP, $\sigma_a$ = 164 MPa, $N_f$ = 1.30 $\times$ 10⁴. (**d**) $\omega$ = 1500 rpm, CJA, $\sigma_a$ = 130 MPa, $N_f$ = 2.14 $\times$ 10⁵. (**e**) $\omega$ = 2500 rpm, CJA, $\sigma_a$ = 126 MPa, $N_f$ = 1.56 $\times$ 10⁵. (**f**) $\omega$ = 2500 rpm, CJA, $\sigma_a$ = 117 MPa, $N_f$ = 2.57 $\times$ 10⁵.

### 3.3. Introduction of Compressive Residual Stress by Cavitation Peening

To examine the cavitation peening effect on residual stress of FSWed AA5754, Figure 11 illustrates the surface residual stress before and after cavitation peening. In Figure 11, cavitation peening was carried out using the cavitating jet in air. Note that the width of the FSWed part at the welded side was about 11 mm. In the case of the welded side, $x = -5.5$ mm—5.5 mm was the FSWed part. As shown in Figure 11, the surface residual stress before peening was tensile at the boundary of FSW on the welded side and at the center on the root side. The values of the tension at the boundary of FSW on the welded side and at the center on the root side were nearly equivalent. These tensile residual stresses are the main reason why the fatigue failure of non-peened specimens occurred at the FSW boundary on the welded side and the root side, as shown in Figure 9. As shown in Figure 11, cavitation peening changed these tensile residual stresses to compression. As shown in Figure 11a,c, the compressive residual stress introduced at the base metal part was larger than that of the FSWed part. This means that cavitation peening can introduce the compressive residual stress into the FSWed area of AA5754, but the introduced compressive residuals stress into the FSWed part by cavitation peening is not as large as that of the base metal part. In any case, cavitation peening can change tensile residual stress to compressive residual stress, and this is the main reason why cavitation peening improves the fatigue properties of FSWed AA5754.

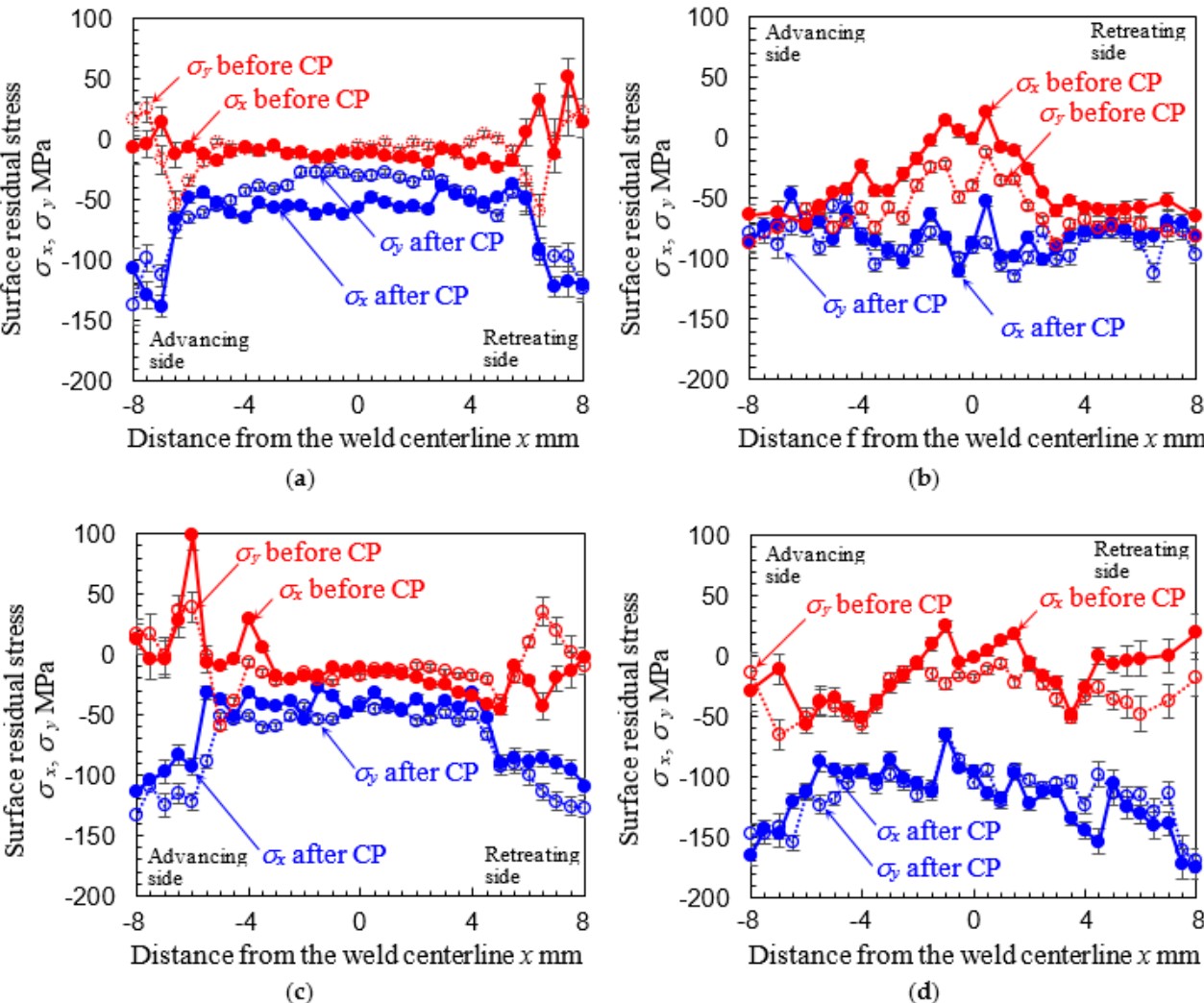

**Figure 11.** Surface residual stress of FSWed AA5754 before and after cavitation peening (CP) measured by the 2D method of X-ray diffraction. (**a**) The welded side of FSW by $\omega$ = 1500 rpm. (**b**) Root side of FSW by $\omega$ = 1500 rpm. (**c**) Welded side of FSW by $\omega$ = 2500 rpm. (**d**) Root side of FSW by $\omega$ = 2500 rpm.

## 4. Conclusions

To demonstrate the effect of cavitation peening on fatigue properties of metal sheet joined by stir friction welding (FSW), FSWed aluminum alloy AA5754 was treated using the cavitating jet in air and water and tested by the displacement-controlled plane-bending fatigue test. To clarify the mechanism of the improvement of the fatigue properties by cavitation peening, the surface residual stress and surface roughness were evaluated. The results obtained can be summarized as follows:

(1) Cavitation peening using a cavitating jet in air (CJA) and a cavitation jet in water (CJW) can improve fatigue properties of FSWed AA5754. The fatigue properties treated by CJA are better than those treated by CJW, as the surface roughness of CJA is smaller than that of CJW.

(2) The fatigue life at $\sigma_a$ = 150 MPa can be more than doubled by CJA and CJW compared with non-peened FSWed specimen. CJA and CJW can also improve the fatigue strength at $N = 10^5$ more than 1.2 times compared with non-peened FSWed specimen.

(3) Cavitation peening using CJA can introduce compressive residual stress into both surfaces of the welded side and the root side of the FSWed part.

(4) In the present FSW condition, the surface residual stress of the FSWed specimen near the center part of the root side and boundary of FSW of the welded side reveals tension.

(5) The fatigue life at $\sigma_a$ = 150 MPa and the fatigue strength at $N = 10^5$ of tested FSW specimen were 3–5% and 53–59% of bulk sheet, respectively. In the present condition, the fatigue properties of FSWed specimen joining at a rotational speed of 1500 rpm were better than those of 2500 rpm.

**Author Contributions:** Conceptualization: H.S. and M.C.; FSW methodology: M.S. and M.C.; peening and fatigue test methodology: H.S.; validation: H.S. and M.C.; formal analysis: H.S.; investigation: H.S.; resources: H.S.; data curation: H.S.; writing—original draft preparation: H.S.; writing—review and editing: H.S., M.S., and M.C.; funding acquisition: H.S. All authors have read and agreed to the published version of the manuscript.

**Funding:** This research was partly supported by JSPS KAKENHI grant numbers 18KK0103 and 20H02021.

**Informed Consent Statement:** Not applicable.

**Data Availability Statement:** The data presented in this study are available on request from the corresponding author.

**Conflicts of Interest:** The authors declare no conflict of interest.

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
