# Peer review of "Effect of Cavitation Peening on Fatigue Properties in Friction Stir Welded Aluminum Alloy AA5754"

_metals, doi:10.3390/met11010059_

Round 1

Reviewer 1 Report

Line 101: Say the number of fatigue specimens you have used with each product (base metal and FSWs)

Line 104: Have you used transversal specimens to determine the ultimate tensile strengths of such FSW welds?

Line 123: v= scanning speed. Do you refer to the sample movement? What is the meaning of the number of the scan,n?

Line 142: Have you used three-point bending fatigue tests? Include the testing span and also the frequency you have applied in these tests

Line 171: Have you ground and polished the sample surface before performing hardness measurements?

Line 189: Fig. 5: Surface damage seems to be much higher after cavitating jet in water. Can you explain this fact?

Line 192: Figure 6: I was not able to see anything important in these images. Can you explain them?, or othewise, discard the figure.

Line 199, Fig. 7: Have you measured them on samples welded with a 2500 or 1500 rotational speed?

Line 208, Fig. 8: Have you measured them on samples welded with a 2500 or 1500 rotational speed?

Line 208, Fig. 8: Cavitation jet peening (in air and also in water) decreases the surface hardness of the FSW parts. Isn´t it unusual? Can you explain this effect?

Line 235, Fig. 9. In the case of the FSW fatigue results, it seems you have used only two experimental fatigue results to determine each fatigue curve. I think you are assuming this is enough, because the fatigue curve may be paralel to the one obtained with the base metal specimens and you interpolate its position with the two experimentally determined points. In fatigue you always have a wide variablity of results and the use of statistical procedures are recommended. More fatigue tests should be performed. Specifically, the fatigue curve after cavitation jet in water (CJW, 2500) is not accurate enough.

Line 259 and Figure 11: Indicate you have measured surface residual stresses.

Fig. 11: You say in the text that these images correspond to cavitating jet in air. Complete the work measuring residual stresses after applying cavitating jet in water.

Reviewer 2 Report

There is not enough data for you to be so specific in your conclusions. Pls "water" them down a tad.

Reviewer 3 Report

In this research the authors test the effect of Cavitation Peening on Fatigue Properties in Friction Stir Welded Aluminum Alloy. 

The research is well presented, the introduction well describe the state of art of FSW, in the materials and methods all the information needful to replicate the experiments are available and the results are well presented. 

The article could be accepted in the present form 

Reviewer 4 Report

Cavitation peening is a derivative of the traditional shot peening which impacts random with a metal ball on the component’s surface at high velocities. This paper applied the cavitation peening to friction stir welding (FSW), a solid‐state joining technique for light‐weight metals and examined the effect of Cavitating Jet in Air(CJA) on number of cycle to failure (Nf) of FSWed aluminum alloy.

Author argues that this article is the first manuscript that has demonstrated the improvement of fatigue properties of FSWed metallic sheet by performing CJA experiment. The structure of this article is acceptable. The reviewer, however, finds the work to be interest. However, the manuscript has several shortcomings. It would be grateful if you would amend the article along the lines suggested. Please prepare a revised version and a list of changes that have been made.

Major revisions

1) In Figure 7: Even though non‐peened specimen (NP) has very low arithmetical mean roughness, compared with CJA, number of cycles to failure of NP (see Fig. 9) is less than FSWed specimens treated by CJA. Because the fatigue failure life is greatly influenced by the surface roughness of the material (specimen), this study shows the opposite result. Detailed explanation is required.

2) In Figure 11: Difference between the residual stresses on the root side of FSW (Fig. 11(b)) and those on the root side of FSW (Fig. 11(d)) rises as rpm increases by 66%. I, however, cannot see difference between the residual stresses on the root side of FSW (Fig. 11(a)) and those on the root side of FSW (Fig. 11(c)). Although the RPM increased, the difference was rather reduced except for the left area (-8nn~-6mm). Is it not normal that the difference in the magnitude of the residual stress generated increases as the RPM increases?

3) In Line 124: It seems to me the combination of parameters (n = 1, v = 4mm/s, tp = 0.25 s/mm) chosen for this works is quite arbitrary. Different results may be produced depending on the combination of the three parameters. A reasonable explanation of the reason for choosing this combination will be very helpful to readers.

4) In Line 136: You must explain why you set the value s/d = 111. This value is so specific that different results may be produced if other value is used.

Minor revisions

1) In Table 1, which test results are compared to ‘CJW at 2500 rpm’.

2) An explanation using figure is necessary so that the readers can understand the terms used in the paper. For example, "Corner of root side" center”, “center part of the root side” and "boundary of welded side".

3) In Line 286: “The fatigue life at ?a = 150 MPa can be more than doubled …” I think it will help readers' study if you explain the specific reason for choosing 150 MPa.
